# Mitigating Class Imbalance in Graph-Structured Data via Hierarchical Learning: Insights from Protein Binding Site Prediction

## Abstract

Learning from imbalanced data remains a major challenge for graph neural networks (GNNs), as minority nodes are not only rare but also structurally marginalized within the graph. We address this issue with CLARA, a hierarchical learning framework that decomposes node classification into two stages: a coarse subgraph-level classifier that selects regions likely to contain minority instances, followed by a fine-grained node-level predictor within these regions. This design improves sensitivity while maintaining scalability, filtering out irrelevant areas and focusing learning on topologically meaningful neighborhoods. Experiments on benchmark graph datasets demonstrate substantial gains over established imbalance-handling methods, with CLARA reaching an F1-score of 88.3%. The same strategy achieves significant improvements in protein–ligand binding site prediction, underscoring its broad and consistent effectiveness across both biological and general graph learning tasks.

## 1 Introduction

Class imbalance is a well-known and persistent challenge in machine learning (ML), where one or more classes are significantly underrepresented (Niaz et al., 2022). Although well-explored in domains like computer vision and natural language processing (Henning et al., 2022; Qu et al., 2025), graph-structured data introduces unique challenges for handling class imbalance: their irregular, non-Euclidean nature and rich relational dependencies often leave minority-class nodes sparse, peripheral, and structurally disadvantaged (Ma et al., 2025; Liu et al., 2025). Graph Neural Networks (GNNs), which rely on message passing across the graph, tend to exacerbate this issue: signals from high-degree, majority-class nodes tend to dominate, overwhelming the representations of minority-class nodes (Xu et al., 2024; Ju et al., 2024). In critical applications of graph-structured data such as disease diagnosis or equipment failure detection, such biases can lead to deceptively high overall accuracy while entirely missing rare but essential cases (Yuan et al., 2022; Pandey et al., 2024; Walke et al., 2024). This problem is particularly pronounced in protein–ligand binding site prediction, where only a small fraction of residues are functionally active, making them difficult to detect amid a structurally dominant background (Xia et al., 2024).

Recent efforts to address class imbalance in graphs can be broadly grouped into three categories: data augmentation (Zhao et al., 2021; Li et al., 2023), loss-function design (Song et al., 2022; Chen et al., 2025), and topological correction (Liu et al., 2021; Zhao et al., 2022; Liu et al., 2023). These approaches have achieved progress by either generating additional minority samples, reweighting losses, or reshaping message passing. However, they typically assume that minority-class nodes are diffusely distributed, overlooking scenarios where minority instances cluster in compact, topologically meaningful subregions, conditions frequently encountered in biological and other scientific networks. In such cases, purely resampling or smoothing strategies may fall short, motivating the need for frameworks that explicitly exploit the localized structure of minority regions.

In this work, we introduce CLARA (Coarse-to-fine Localized Adaptive Region Attention), a hierarchical learning framework for addressing class imbalance in graphs. CLARA progressively refines node classification through two stages: a coarse subgraph-level classifier first isolates regions enriched with minority nodes, and a fine-grained node-level predictor then operates within these re-

gions. This design enhances sensitivity to rare classes while preserving scalability, as computation is shifted from the entire graph to localized substructures. A further strength of CLARA lies in its heuristic-based subgraph modeling, which allows domain knowledge, such as solvent-accessible surface area in proteins, to guide region selection and improve the quality of candidate subgraphs. While the framework is broadly applicable across domains, it is originally inspired by biological settings such as protein–ligand binding sites, where minority residues naturally cluster into compact, functionally meaningful regions.

To demonstrate the effectiveness and generality of our approach, we focus on protein–ligand binding site prediction, an emblematic case of class imbalance in structured biological data. Identifying these sites is a fundamental task in structural bioinformatics, with implications for understanding protein function, virtual screening, and drug design (Konc & Janežič, 2014; Boike et al., 2022; Che et al., 2024). Binding residues typically represent less than 6% of all amino acids, forming compact and functionally critical regions within the protein structure (Kulandaisamy et al., 2017). This makes the task particularly challenging for learning algorithms, especially those sensitive to data imbalance. By modeling proteins as residue-level graphs with spatially defined edges, we naturally capture the structural context of binding sites while exposing the inherent sparsity and imbalance that characterize the task. We evaluate CLARA not only on biologically grounded datasets for protein–ligand binding site prediction but also on widely used graph benchmarks under controlled imbalance settings. This dual evaluation confirms the relevance of our method across both application-driven and general-purpose scenarios.

The main contributions of our work are: *(i)* introducing a subgraph-based modeling strategy that decomposes large graphs into localized neighborhoods, improving scalability and reducing noise; *(ii)* proposing a hierarchical coarse-to-fine learning pipeline that concentrates prediction capacity on structurally informative regions, enhancing minority-class sensitivity; and *(iii)* incorporating heuristic-based subgraph modeling, such as solvent accessibility in proteins, while remaining flexible enough to integrate diverse criteria in other domains. Together, these elements provide a scalable framework particularly well suited for scenarios where minority classes are concentrated in localized graph regions, yielding state-of-the-art results with up to 88.3% macro-F1 on benchmark graph datasets and 68.6% F1-score in protein–ligand binding site prediction.

## 2 RELATED WORKS

Recent progress in machine learning and Graph Neural Networks (GNNs) has enabled advances across domains like chemistry, social networks, and biology (Wittmann et al., 2021; Kedi et al., 2024; Sharma et al., 2024; Abadal et al., 2021). However, real-world graphs often present challenges such as skewed label distributions and irregular topologies. This section reviews two research directions central to our work: approaches for handling class imbalance in graphs and methods for protein binding site prediction, the primary application of our proposed framework.

**Class Imbalance in Graph Learning.** Addressing class imbalance in graph-based learning has become increasingly important, as graph data presents unique challenges not encountered in traditional settings. Node interdependence, structural irregularities, and non-Euclidean topology can amplify the effects of skewed label distributions, making conventional solutions less effective (Liu et al., 2025). To mitigate these issues, various methods have been proposed, including resampling techniques, loss function engineering, and graph-specific architectural adaptations (Haixiang et al., 2017; Ma et al., 2025). Below, we outline key strategies developed for node classification under imbalance and discuss their limitations, which motivate the structured design of our hierarchical learning framework.

Recent surveys on class-imbalanced learning in graphs highlight the unique challenges posed by graph-structured data, including bias propagation through message passing, structural imbalance, and the difficulty of applying traditional techniques like oversampling, without disrupting topology (Ma et al., 2025; Ju et al., 2024; Liu et al., 2025). These works propose taxonomies that categorize methods by problem type (e.g., node vs. graph classification, class vs. structure imbalance) and strategy (e.g., cost-sensitive learning, topology-aware sampling), while also emphasizing the need for context-aware, generalizable models. Notably, they underscore the importance of developing graph-specific solutions, like our hierarchical approach, that can simultaneously handle data imbalance and structural sparsity in a principled manner.

Approaches to address class imbalance in graphs generally fall into three main categories: data augmentation, loss function design, and topological correction. Data augmentation methods focus on over-sampling the minority class. GraphSMOTE (Zhao et al., 2021) synthesizes new nodes in the learned embedding space while generating edges that preserve local structure. GraphSHA (Li et al., 2023) enhances minority-class representations by creating harder examples and employing a SemiMixup module to balance informativeness and stability. Loss function design offers an alternative route. TAM (Song et al., 2022) adapts decision boundaries using local topological cues, while NodeImport (Chen et al., 2025) dynamically reweights training based on node importance, prioritizing those that contribute most to generalization. Topological correction strategies explicitly target structural imbalance: Tail-GNN (Liu et al., 2021) transfers information from high to low-degree nodes, and TopoImb (Zhao et al., 2022) leverages motif-aware regularization. BAT (Biased Attention Transformer) (Liu et al., 2023) proposes a lightweight topological augmentation framework that mitigates predictive bias by correcting structural deficiencies without explicit class reweighting.

Collectively, these works highlight that addressing class imbalance in graphs requires not only data augmentation or loss design, but also architectural and topological adaptations tailored to graph structure. Nonetheless, most approaches still struggle when minority instances are highly localized and structurally concentrated, as often observed in biological networks. Recently, complementary directions have emerged that exploit large language models (LLMs), such as LA-TAG (Wang et al., 2024), which generates synthetic minority nodes in text-attributed graphs using LLM-based augmentation. While promising in settings with rich semantic information, these methods remain largely unexplored for structural domains, underscoring the need for frameworks like CLARA that directly exploit the clustered nature of minority instances.

**Protein Binding Site Prediction.** Identifying ligand-binding residues is a foundational task in structural bioinformatics, underpinning efforts in drug discovery and protein function analysis (Konc & Janežič, 2014; Boike et al., 2022). The task involves identifying residues within a protein structure that are responsible for interacting with ligands or other biomolecules. Over the years, a variety of computational methods have been developed to automate this process, ranging from geometry-based heuristics to machine learning models trained on annotated datasets (Zhao et al., 2020). More recently, deep learning (DL) approaches have leveraged structural and sequence-based representations to improve prediction accuracy (Rohulia et al., 2025). Following, we highlight representative tools and models that have shaped the field and serve as comparative baselines in our evaluation.

Recent reviews have provided comprehensive insights into protein-ligand binding site (LBS) prediction from different angles. Xia et al. (2024) provides an in-depth overview of protein-ligand binding site prediction, highlighting its critical role in protein function annotation and drug discovery. Dhakal et al. (2022) explore the role of artificial intelligence in protein-ligand interaction prediction, reviewing ML-based techniques for predicting binding sites, affinities, and poses. The authors note that binding site prediction suffers from severe class imbalance, which challenges conventional ML algorithms. To address this, ensemble methods with random undersampling have been proposed. Moreover, attention-based neural architectures, proven effective in protein structure prediction, are identified as promising alternatives to traditional convolutional or recurrent models. The survey advocates for multi-task frameworks capable of jointly modeling site, affinity, and pose predictions to better exploit their interdependencies. A recent review by Rohulia et al. (2025) surveys state-of-the-art deep learning methods for binding site prediction, covering model architectures, data resources, and evaluation practices. The work emphasizes the role of CNNs and GNNs in advancing predictive accuracy and highlights how curated databases underpin the development of these models.

A wide range of methods has been developed for ligand binding site prediction, from classical heuristic-based tools to modern deep learning models. Traditional approaches like COACH (Yang et al., 2013) combine sequence and structural similarity in a meta-predictor framework, while Fpocket (Le Guilloux et al., 2009) uses geometric criteria to efficiently detect cavities, recovering the majority of known sites. GRaSP (Santana et al., 2022), in turn, applies a residue-centric graph-based strategy, achieving strong performance with reduced runtime. More recent machine learning methods include P2Rank (Polák et al., 2025), which predicts ligandability scores from local features without templates, and PUResNetV2.0 (Jeevan et al., 2024), a deep model combining U-Net and ResNet modules with sparse representations. LigBind (Xia et al., 2023) further advances the field by incorporating ligand-specific context through relation-aware graph neural networks. Together, these methods reflect the shift toward scalable, structure-informed, and learning-based approaches for accurate binding site identification. In this context, our method CLARA also operates as a bind-

ing site predictor, but distinguishes itself by explicitly addressing the severe class imbalance of the task through hierarchical subgraph-based learning, thereby improving sensitivity to the minority residues that define functional binding regions.

More recently, large language models (LLMs) have opened new perspectives for binding site prediction by leveraging sequence-based representations. For example, recent work demonstrated that unsupervised LLMs trained on enzyme–substrate reactions can recover more than 50% of binding site residues directly from sequence data, capturing signals of substrate recognition and atomic-level interactions without explicit structural supervision (Teukam et al., 2024). Similarly, protein language models such as ESM and ESMFold (Lin et al., 2023) have shown that scaling transformer architectures to billions of parameters enables accurate atomic-level structure prediction directly from single sequences, bypassing the need for multiple sequence alignments. While these advances highlight the potential of LLMs to encode structural determinants of binding sites, they do not directly address the extreme class imbalance inherent to residue-level prediction. CLARA complements these directions by introducing a scalable graph-based framework tailored to scenarios where binding residues occur as compact and highly localized minority regions.

## 3 METHODOLOGY

The problem of class imbalance arises when certain classes, typically those of greatest interest, are underrepresented compared to others in the training data. This issue is widespread across domains such as disease diagnosis, fraud detection, cybersecurity, and rare event modeling (Wheelus et al., 2018; Yuan et al., 2022). Imbalance can be either *intrinsic*, stemming from natural distributions (e.g., most patients are healthy), or *extrinsic*, introduced through biased data collection (Wang et al., 2021). Regardless of its origin, imbalance compromises the performance of standard machine learning models, especially in complex tasks (Niaz et al., 2022). Traditional metrics like accuracy are misleading in this context, as models may achieve high scores while completely ignoring the minority class (Japkowicz, 2013). A common measure of imbalance severity is the *imbalance ratio*, defined as $\max_k |C_k| / \min_k |C_k|$, where $|C_k|$ denotes the number of samples in class $k$. In the following, we first present our general hierarchical framework for learning under class imbalance in graphs, and in Section 3.1 we detail its application to the task of protein binding site prediction.

Graphs provide a natural representation for relational data, where a graph $G = (V, E)$ is defined by a set of nodes $V$ and edges $E$ that encode relationships between entities. In this work, we focus on node classification, where the goal is to assign a label to each node in the graph. Graph Neural Networks (GNNs) address this task by propagating information through message passing, learning embeddings that integrate both node features and topological context (Huang et al., 2022). However, under class imbalance, GNNs face unique challenges: minority-class nodes are not only rare but also structurally marginalized, often appearing in peripheral or sparsely connected regions (Ju et al., 2024). This topological isolation reduces the effectiveness of message passing and biases learned representations toward majority classes (Liu et al., 2025).

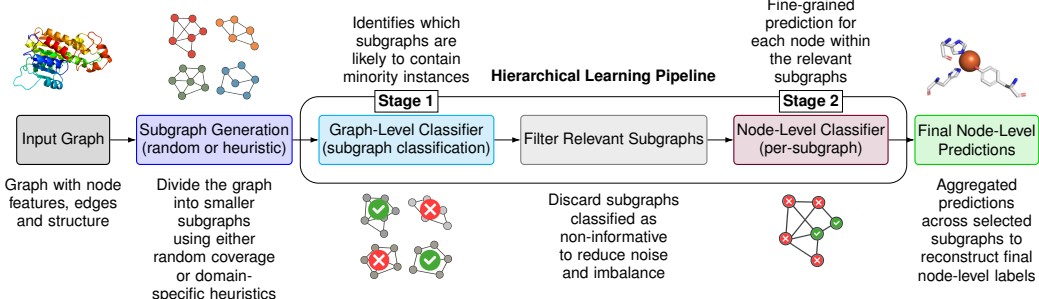

Figure 1: Schematic of the hierarchical framework for addressing class imbalance in graphs. The pipeline combines subgraph decomposition, graph-level filtering, and node-level classification to focus learning on structurally relevant regions.

To address these challenges, we introduce CLARA (Coarse-to-fine Localized Adaptive Region Attention), a hierarchical learning framework tailored to class-imbalanced node classification. CLARA consists of two stages: *(i)* a coarse subgraph-level classifier that identifies regions likely to contain minority nodes, and *(ii)* a fine-grained node-level classifier applied within these regions. This coarse-to-fine strategy progressively narrows the model's focus, improving sensitivity to underrepresented classes while reducing noise from majority-dominated regions. An overview of the pipeline is shown in Figure 1.

**Subgraph Construction.** Given an input graph $G = (V, E)$, we construct a collection of subgraphs $G_i = (V_i, E_i)_{i=1}^N$ by selecting root nodes $r_i \in V$ and extracting their $k$-hop neighborhoods. The choice of root nodes determines both the number and the quality of subgraphs. In principle, any root selection policy can be adopted; here, we consider two strategies: *(i) Randomized coverage via node coloring.* Root nodes are sampled uniformly at random from the set of unassigned nodes. For each root $r_i$, the $k$-hop neighborhood $G_i$ is extracted, and all nodes in $V_i$ are marked as "colored." Colored nodes are not eligible to be selected as roots in subsequent iterations, though they may still appear as neighbors in other subgraphs. This process continues until no uncolored nodes remain or a stopping criterion is met. Compared to exhaustive per-node extraction, which generates $|V|$ overlapping neighborhoods, this randomized coverage produces a substantially smaller set of subgraphs, while still achieving near-complete coverage with reduced redundancy. *(ii) Domain-guided selection.* In scenarios where domain knowledge is available, root nodes can be selected according to heuristic scores $s : V \to \mathbb{R}$ that reflect task-specific relevance. For example, in protein graphs, residues with solvent accessible surface area (SASA) (Hubbard & Thornton, 1993) above a threshold $\tau$ are chosen as roots, i.e., $R = v \in V \mid s(v) > \tau$. Subgraphs are then constructed as $k$-hop neighborhoods around $R$. This approach reduces randomness and concentrates computation on structurally or functionally meaningful regions, often improving the quality of downstream predictions. Together, these strategies illustrate the trade-off between scalability and task specificity: randomized coverage provides efficient, domain-independent decomposition, whereas heuristic-based selection integrates prior knowledge to bias the model toward more informative subregions.

**Stage 1: Subgraph-Level Classification.** The first classifier operates at the subgraph level and serves as a coarse filter. Each subgraph $G_i = (V_i, E_i)$ is encoded by a graph neural network $f_{\text{sub}}$, producing node embeddings that are aggregated into a fixed-length vector $\mathbf{h}_i = \rho(f_{\text{sub}}(G_i)) \in \mathbb{R}^d$, where $\rho$ is a readout function (e.g., mean, sum, or attention pooling) that maps variable-sized subgraphs to fixed-dimensional representations. A binary classifier $\phi_{\text{sub}} : \mathbb{R}^d \to \{0, 1\}$ then predicts $\hat{y}_i = \phi_{\text{sub}}(\mathbf{h}_i)$, where $\hat{y}_i = 1$ indicates that $G_i$ likely contains at least one minority-class node. This model is trained independently with ground-truth labels at the subgraph level. At inference time, it is applied first: subgraphs with $\hat{y}_i = 0$ are pruned, and only those with $\hat{y}_i = 1$ are forwarded to Stage 2. This staged inference is crucial under severe class imbalance, since it prevents the node-level classifier from being overwhelmed by majority-dominated regions. While the two classifiers are trained separately, their sequential application during inference forms the hierarchical pipeline. From a computational standpoint, this reduces noise, increases efficiency, and enriches the effective concentration of minority-class nodes in the subgraphs that reach Stage 2.

**Stage 2: Node-Level Classification.** The second stage performs fine-grained classification within the set of subgraphs $\mathcal{S}$ identified by Stage 1 during inference. For each subgraph $G_i = (V_i, E_i) \in \mathcal{S}$, a node-level encoder $f_{\text{node}}$ produces embeddings $\mathbf{H}_i = f_{\text{node}}(G_i) \in \mathbb{R}^{|V_i| \times d}$, where each row corresponds to a node embedding $\mathbf{h}_j$. A node-wise classifier $\phi_{\text{node}} : \mathbb{R}^d \to \{0, 1\}$ then assigns labels $\hat{y}_j = \phi_{\text{node}}(\mathbf{h}_j)$ for nodes $v_j \in V_i$. The final prediction for each node in the original graph is obtained by merging predictions across all retained subgraphs: a node is labeled positive if it is predicted as such in at least one subgraph, while nodes absent from all positively classified subgraphs are implicitly labeled negative. During training, $f_{\text{node}}$ and $\phi_{\text{node}}$ are optimized using all available subgraphs, with supervision applied directly at the node level. At inference time, however, only subgraphs selected by Stage 1 are processed, which reduces noise and improves computational efficiency. This design yields two main benefits: *(i)* restricting inference to structurally relevant subgraphs improves the effective class balance presented to the node-level classifier, and *(ii)* attention mechanisms in $f_{\text{node}}$ allow the model to prioritize nodes that contribute most to minority-class discrimination, mitigating the dilution of minority signals present in global training.

## 3.1 Hierarchical Learning for Protein Binding Site Prediction

In this subsection, we describe how the proposed hierarchical learning framework is instantiated for binding site prediction. We outline the main components of the pipeline: *(i)* graph modeling, where protein structures are represented as residue-level graphs; *(ii)* subgraph construction, using either randomized or heuristic-based strategies; and *(iii)* the two-stage classification process, combining coarse subgraph-level filtering with fine-grained node-level prediction.

**Problem Definition.** Ligand binding site prediction aims to identify which amino acid residues in a protein are directly involved in interactions with small molecules (Chen et al., 2011). From a bioinformatics perspective, these residues are of central functional importance, as they mediate molecular recognition and drug binding (Zhao et al., 2020). Computationally, however, the task is highly imbalanced: binding residues typically constitute less than 6% of all residues in a protein, with the vast majority belonging to the non-binding class (Kulandaisamy et al., 2017). This imbalance poses two main challenges. First, the scarcity of positive residues makes it difficult for learning algorithms to detect the subtle patterns that distinguish them from the background. Second, binding residues are not randomly distributed; rather, they form compact structural clusters on the protein surface. Any effective method must therefore handle both the label imbalance and the topological concentration of positives within localized regions of the protein graph.

**Graph Modeling.** To model protein structures as graphs, we represent each residue as a node, and non-covalent interactions between residues as edges. The presence of an edge between two residues is determined by whether there exists at least one non-covalent interaction between them. These interactions are computed based on interatomic distances and atom types, using a kd-tree-based implementation provided by Biopython (Cock et al., 2009), which enables efficient identification of interactions without the need to calculate all pairwise distances. The types of interactions considered include aromatic stacking, disulfide bridges, hydrogen bonds, hydrophobic contacts, salt bridges, and repulsive forces. We do not use edge features in our model. Instead, each node is represented by a high-dimensional feature vector obtained from the ESM-2 model (Evolutionary Scale Modeling), a pretrained protein language model that captures rich biological information based on protein sequence (Lin et al., 2023).

**Data Preparation.** For the binding site prediction task, we used curated protein–ligand interaction data from BioLiP (Zhang et al., 2024), where residues are labeled as binding or non-binding. Only proteins with complete 3D coordinates were retained to ensure structural consistency. Proteins were then decomposed into local subgraphs capturing residue neighborhoods. We applied two strategies: *(i)* randomized coverage with node coloring, which partitions the protein graph into $k$-hop neighborhoods while reducing redundancy compared to exhaustive per-node extraction; and *(ii)* a heuristic approach based on solvent-accessible surface area (SASA), where surface-exposed residues are chosen as subgraph roots. Dataset statistics and implementation thresholds are provided in Appendix A.

**Hierarchical Classification.** After subgraph generation, classification proceeds through the two-stage hierarchical framework described in Section 3. In Stage 1, a subgraph-level classifier determines whether each subgraph is likely to contain positive residues. In Stage 2, a node-level classifier refines predictions within the retained subgraphs, and outputs are merged to obtain residue-level labels for the full protein. Both classifiers are implemented with Graph Attention Networks (GATs), chosen for their ability to adaptively weight neighbors during message passing. Additional architectural and training details are reported in Appendix B.

## 4 Results

In this section, we evaluate the proposed hierarchical learning framework in the context of graph imbalance, comparing against state-of-the-art methods designed to address class imbalance in graph-structured data. We first present experiments on the Planetoid datasets (Cora, CiteSeer, PubMed) (Yang et al., 2016), widely used benchmarks for node classification under varying degrees of imbalance, where we contrast our approach with representative imbalance-handling strategies. We then turn to ligand binding site prediction, comparing against specialized structure-based predictors.

## 4.1 EVALUATION ON CLASS-IMBALANCED GRAPHS

To evaluate the effectiveness of our framework under controlled imbalance, we conducted experiments on the Planetoid datasets Cora, CiteSeer, and PubMed (Yang et al., 2016), each widely used in graph learning research. Cora contains 2,708 nodes and 5,429 edges, CiteSeer 3,327 nodes and 4,732 edges, and PubMed 19,717 nodes and 44,338 edges. All experiments followed the public splits with an imbalance ratio of 10. Since the task was multi-class node classification, we adopted a one-vs-rest (OvR) transformation and report *macro-F1*, which equally weights all classes and thus balances precision and recall. This choice is particularly important in class-imbalanced scenarios: unlike micro-F1 or accuracy, which can be dominated by majority classes, macro-F1 highlights the model's ability to correctly identify minority classes, making it a more reliable indicator of overall performance.

We compared two variants of our framework. (*i*) **CLARA-S**, a simplified version that uses subgraph decomposition but omits hierarchical filtering, applying node-level classification to all generated subgraphs. (*ii*) **CLARA**, the full hierarchical model, where Stage 1 filters candidate subgraphs and Stage 2 performs fine-grained node classification. In both cases, subgraphs were generated using the coloring-based strategy described in Section 3. As baselines, we included three imbalance-handling methods: NodeImport (Chen et al., 2025), BAT (Liu et al., 2023), and GraphSHA (Li et al., 2023). NodeImport dynamically reweights the loss to emphasize minority-class nodes, BAT introduces topology-sensitive augmentation through biased attention, and GraphSHA enhances minority representations via semi-supervised hard example generation. All models, including our variants, were implemented with Graph Convolutional Networks (GCNs) as the backbone. Figure 2 reports macro-F1 (%) as mean values with 99% confidence intervals across 10 runs.

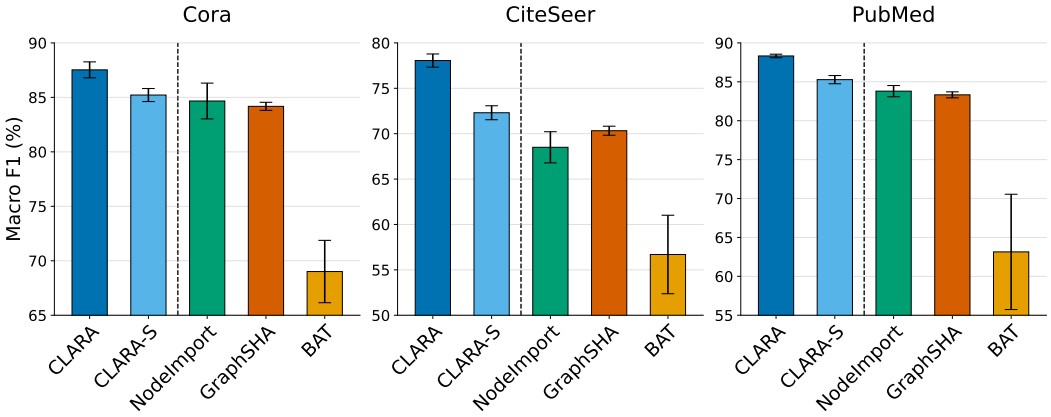

Figure 2: Macro-F1 (%) with 99% confidence intervals on the Cora, CiteSeer, and PubMed datasets. Results are reported for CLARA, CLARA-S, NodeImport, GraphSHA, and BAT.

The results consistently favor our hierarchical design. In the Cora dataset, CLARA achieved an average macro-F1 of 87.6%, compared to 85.3% for CLARA-S, 84.6% for NodeImport, and 84.1% for GraphSHA. BAT lagged substantially at 68.3%, showing that biased attention alone is insufficient under strong imbalance. In the CiteSeer dataset, the gap widened: CLARA obtained 78.0%, while CLARA-S reached 72.7%, NodeImport 68.9%, and GraphSHA 70.6%. BAT again performed poorly, averaging only 57.0%. Finally, in the PubMed dataset, the largest and most structurally diverse dataset, CLARA delivered 88.3%, outperforming CLARA-S (85.4%), NodeImport (83.8%), and GraphSHA (83.3%). Importantly, the 99% confidence intervals for CLARA do not overlap with those of any competing method, confirming the statistical significance of its superiority across all datasets.

A deeper analysis highlights several insights. First, even without hierarchy, CLARA-S already surpasses all imbalance-aware baselines, showing that subgraph decomposition alone is a strong strategy for reducing topological noise and improving the local balance of minority instances. However, CLARA consistently yields an additional performance margin: the hierarchical filter discards irrel-

evant or majority-dominated subgraphs, concentrating learning on regions more likely to contain minority nodes. This coarse-to-fine decomposition enables complementary signals, subgraph-level evidence followed by node-level refinement, that neither subgraphs alone nor traditional imbalance mitigation methods can fully capture.

Second, the relative margins vary with dataset size and structure. On CiteSeer, where class separation is weaker and imbalance effects are stronger, CLARA's advantage over CLARA-S is more pronounced (a +5.3% F1 improvement). On PubMed, the gain is smaller in absolute terms (+2.9%) but still statistically significant, showing that hierarchical filtering remains beneficial even on larger graphs with richer connectivity. On Cora, the performance gap is moderate (+2.3%), suggesting that hierarchical benefits are consistent but adapt to the underlying data structure. Together, these patterns show that CLARA generalizes well across graph domains, scaling from small citation networks to larger text corpora.

### 4.2 BENCHMARK EVALUATION ON BINDING SITE PREDICTION.

After establishing the effectiveness of our framework on controlled benchmarks, we next evaluate its performance in a biologically grounded setting. Evaluation was conducted on the COACH100 dataset (Jeevan et al., 2024), which consists of 65 non-redundant proteins annotated with 100 curated ligand-binding sites and is widely used to benchmark binding site predictors under realistic yet relatively simple structural conditions.

For this evaluation, we considered four variants of our method: (*i*) **Flat GNN**, a baseline trained directly on the full residue-level graph without decomposition; (*ii*) **CLARA-S**, which applies node-level classification on all subgraphs without hierarchical filtering; (*iii*) **CLARA**, our hierarchical model using randomized coloring for subgraph generation; and (*iv*) **CLARA-SASA**, which employs solvent-accessible surface area (SASA) as a heuristic for subgraph generation. All four approaches were implemented using Graph Attention Networks (GATs) as the backbone architecture, and relied on the same graph modeling pipeline described in Section 3.1, including residue-level node features derived from ESM-2 embeddings, interaction-based edge definitions, and identical training procedures. Further dataset statistics, preprocessing details, and implementation parameters are provided in Appendix X. Table 1 summarizes results in terms of Matthews Correlation Coefficient (MCC), precision, recall, and F1-score.

Table 1: Performance comparison on the COACH100 dataset. All methods use the same graph modeling pipeline (Section 3.1). Results are reported for four configurations of our framework: Flat GNN (no subgraph decomposition), CLARA-S (subgraphs only), CLARA (hierarchical with randomized coloring), and CLARA-SASA (hierarchical with solvent-accessible surface area heuristic subgraph generation). Competing baselines GRaSP, PUResNet v2.0, NodeImport, BAT, and GraphSHA are also shown for reference.

| Method | MCC | Precision | Recall | F1-score |
|---|---|---|---|---|
| GRaSP | 0.501 | 63.5% | 43.9% | 51.9% |
| PUResNet v2.0 | 0.624 | 61.3% | 62.4% | 61.8% |
| BAT | 0.630 | 62.5% | **67.5%** | 64.9% |
| GraphSHA | 0.632 | 63.0% | 62.8% | 65.1% |
| NodeImport | 0.637 | 66.1% | 65.0% | 65.5% |
| Flat GNN | 0.618 | 56.1% | 66.2% | 62.4% |
| CLARA-S | 0.653 | **76.6%** | 60.0% | 66.5% |
| CLARA | 0.660 | 72.0% | 63.4% | 67.4% |
| CLARA-SASA | **0.674** | 75.0% | 63.2% | **68.6%** |

**Comparison with Existing Predictors.** We first compared against PUResNet v2.0 (Jeevan et al., 2024), a point-cloud deep learning model, and GRaSP (Santana et al., 2022), a graph-based predictor retrained on local neighborhoods. Both represent strong baselines in the field of binding site prediction, but neither relies on our residue-level graph modeling. As a result, they fall behind even the simplest variant of our framework. The Flat GNN, trained directly on full protein graphs, achieved an MCC of 0.618 and an F1-score of 62.4%, surpassing PUResNet (MCC 0.624, F1 61.8%) and sub-

stantially outperforming GRaSP (MCC 0.501, F1 51.9%). This result underscores two points. First, residue-level graph modeling (Section 3.1) is by itself a strong foundation, capturing dependencies that geometric or neighborhood-only methods overlook. Second, our framework does not merely introduce a hierarchical pipeline for imbalance handling but also builds on a modeling strategy that already improves binding site prediction in its own right.

**Comparison with Graph-Based Imbalance Methods.** We then evaluated methods designed explicitly to handle class imbalance: NodeImport (Chen et al., 2025), BAT (Liu et al., 2023), and GraphSHA (Li et al., 2023). All three outperform GRaSP, PUResNet, and also the Flat GNN, confirming that imbalance-aware training provides measurable benefits in this setting. NodeImport reached an MCC of 0.637 and an F1-score of 65.5%, offering the most balanced trade-off. BAT achieved the highest recall (67.5%), but at the cost of lower precision, while GraphSHA yielded competitive precision and recall, translating into an F1-score of 65.1%. These results highlight that reweighting, attention bias, and oversampling indeed strengthen predictions under imbalance. However, they still treat the problem primarily at the level of sample distribution, without exploiting the structural concentration of binding residues. In contrast, our framework leverages both modeling and hierarchical filtering to focus directly on regions where the minority class is most likely to occur.

**Impact of Hierarchical Learning.** Finally, we compared the four variants within our own framework: Flat GNN, CLARA-S, CLARA, and CLARA-SASA. This analysis also serves as an ablation study. The Flat GNN establishes a competitive baseline but suffers from precision loss (56.1%) due to majority dominance, despite recall being relatively high (66.2%). CLARA-S alleviates this by decomposing graphs into subgraphs, raising MCC to 0.653 and F1-score to 66.5%, which confirms the value of localized decomposition in balancing minority visibility. CLARA further improves by filtering out irrelevant subgraphs: its hierarchical design increases precision to 72.0% and achieves an F1-score of 67.4%. The best results come from CLARA-SASA, which incorporates solvent accessibility as a heuristic for subgraph generation, reaching the highest MCC (0.674) and F1-score (68.6%). This progression shows that decomposition improves local balance, hierarchy reduces structural noise, and heuristics amplify these benefits by guiding the model toward functionally relevant regions. Together, these components establish CLARA-SASA as the most effective approach, outperforming not only general imbalance methods but also domain-specific predictors.

## 5 CONCLUSION

We introduced CLARA, a hierarchical learning framework that addresses class imbalance in graphs by decomposing node classification into two stages: a coarse subgraph-level classifier that filters structurally relevant regions, followed by a fine-grained node-level predictor. This design improves sensitivity to minority classes, reduces noise, and maintains scalability by shifting computation from full graphs to localized substructures. On the COACH100 benchmark for protein–ligand binding site prediction, CLARA with solvent-accessibility–based subgraph selection achieved an MCC of 0.674 and an F1-score of 68.6%, surpassing state-of-the-art predictors. Significant gains were also observed on general graph benchmarks, with CLARA reaching 88.3% macro-F1 on PubMed, demonstrating its effectiveness beyond biological applications. A key strength of CLARA lies in its ability to leverage heuristic-based subgraph modeling, which allows domain knowledge to guide region selection, and in its suitability for scenarios where minority instances are concentrated in specific graph regions rather than uniformly distributed. Comparisons with imbalance-handling methods such as NodeImport, BAT, and GraphSHA confirm that while these approaches provide meaningful improvements, they fall short of the systematic benefits achieved by our hierarchical framework. Overall, CLARA provides a scalable and general solution for class-imbalanced graph learning, with broad applicability from molecular biology to diverse graph-structured domains.

## REPRODUCIBILITY STATEMENT

We have taken several measures to ensure the reproducibility of our work. All datasets used are publicly available: Planetoid benchmarks (Cora, CiteSeer, PubMed) and the COACH100 dataset for protein–ligand binding site prediction. The procedures for graph construction, subgraph generation (coloring and heuristic-based strategies), and model architectures are detailed in Sections 3 and 3.1.

Hyperparameters, training protocols, and additional implementation details are provided in Appendices A and B. Full experimental results, including confidence intervals and mean values across multiple runs, are reported in Sections 4.1 and 4.2, ensuring that our findings can be independently verified and reproduced.

## DECLARATION OF GENERATIVE AI AND AI-ASSISTED TECHNOLOGIES IN THE WRITING PROCESS

During the preparation of this manuscript, the authors used AI-assisted tools (Grammarly and Chat-GPT) exclusively for improving language clarity, grammar, and readability. No part of the research design, data analysis, or scientific contributions relied on these tools. After their use, the authors carefully reviewed and revised the text, and they take full responsibility for the final content of the paper.

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

# A  DATA PREPARATION DETAILS

## A.1  DATASET CONSTRUCTION

The dataset used for training and evaluation was derived from the BioLiP database (Zhang et al., 2024), which provides curated annotations of biologically relevant ligand–protein interactions. Each residue in BioLiP is explicitly labeled as binding or non-binding, enabling supervised learning formulations. To avoid redundancy and ensure structural consistency, two filters were applied: (i) only non-redundant protein entries were retained, and (ii) only structures with complete 3D atomic coordinates available in the Protein Data Bank (PDB) were considered. After preprocessing, the resulting dataset comprised 71,305 annotated binding residues distributed across 32,209 unique proteins. This large-scale set served as the pool of training templates for our method.

## A.2  DYNAMIC TRAINING REGIME

Instead of training a single static model, we adopt a dynamic training paradigm inspired by the GRaSP strategy (Santana et al., 2020). For each query protein, homologous proteins are retrieved from the training set using BLAST sequence alignment. Up to 30 homologs are retained, and the model is retrained on-the-fly using this subset, producing an instance-specific predictor. This design ensures that the model specializes according to the structural and evolutionary context of each query protein, while avoiding overfitting to unrelated proteins. In practice, each prediction benefits from its own fine-tuned model, rather than relying on a single global model trained on all proteins.

## A.3  SUBGRAPH GENERATION STRATEGIES

To represent proteins as graphs, we decompose each structure into localized subgraphs that encode residue neighborhoods. Two complementary strategies were explored:

- **Randomized coverage with node coloring.** Subgraphs are generated iteratively to ensure that all residues are covered while minimizing redundancy. A residue is randomly selected as the root, and its $k$-hop neighborhood (with $k = 3$) is extracted to form a subgraph. All nodes included in this subgraph are marked as "colored," meaning they will not be reused as roots in subsequent iterations. This procedure continues until all residues are covered. Compared to the naive approach of generating a subgraph for every residue, this method reduces the total number of subgraphs by approximately 80% while retaining essentially the same structural information.

- **Heuristic approach (SASA-based).** As an alternative, subgraphs can be seeded using biologically motivated heuristics. Specifically, solvent accessibility was used as a criterion, computed with NACCESS (Hubbard & Thornton, 1993). Residues with Solvent Accessible Surface Area (SASA) greater than 65% of their theoretical maximum were designated as candidate roots, and their 3-hop neighborhoods were extracted as subgraphs. Unlike the randomized coverage strategy, no coloring was enforced, allowing overlaps when exposed residues are spatially close. This redundancy is intentional, as solvent-exposed residues are more likely to participate in ligand binding and may benefit from multiple overlapping contexts.

## B  HIERARCHICAL CLASSIFICATION DETAILS

### B.1  CLASSIFIER DESIGN

The hierarchical framework consists of two classifiers implemented with Graph Attention Networks (GATs). GATs were selected over alternatives such as GCNs because their attention mechanism allows the model to dynamically prioritize neighbors during message passing, which is particularly relevant in protein graphs where neighboring residues may differ widely in functional importance. Importantly, the two classifiers are trained independently: Stage 1 is optimized to classify subgraphs, while Stage 2 is optimized to classify residues. The hierarchy emerges only at inference time, when the predictions are combined.

### B.2  STAGE 1: SUBGRAPH-LEVEL CLASSIFICATION

The first stage performs graph-level binary classification, predicting whether each subgraph contains at least one binding residue. Subgraphs predicted as negative are discarded before Stage 2. Subgraph embeddings are obtained by passing node features through stacked GAT layers, followed by batch normalization. A global AddPooling layer aggregates the node embeddings into a fixed-length vector. Other pooling operators (mean, attention-based) were also tested, but AddPooling consistently provided a balance between simplicity and effectiveness. During training, ground-truth labels are used, while at inference Stage 1 predictions determine which subgraphs advance to Stage 2.

### B.3  STAGE 2: NODE-LEVEL CLASSIFICATION

The second stage applies residue-level classification within the set of subgraphs retained from Stage 1. Node embeddings are computed with GAT layers, and each residue is classified independently. Since this stage operates only within filtered subgraphs at inference time, it benefits from a higher local proportion of positives compared to training on the full protein graph. Attention weights further allow the model to focus on residues most informative for binding. Importantly, during training, Stage 2 uses all subgraphs (positive and negative), whereas at inference only the subgraphs predicted as positive by Stage 1 are processed.

### B.4  AGGREGATION OF PREDICTIONS

Final residue-level predictions are obtained by merging node-level outputs across all subgraphs that passed Stage 1. A residue is labeled as positive if it is predicted as positive in at least one subgraph; residues absent from all positive subgraphs are labeled negative. This aggregation strategy balances precision and recall: filtering reduces noise, while merging ensures coverage even when subgraphs overlap.

### B.5  TRAINING DETAILS

Both classifiers use two GAT layers with hidden dimension 128, ReLU activation, dropout of 0.3, and batch normalization. The Adam optimizer is used with an initial learning rate of $10^{-3}$ and weight decay of $5 \times 10^{-4}$. Training is performed with class-weighted binary cross-entropy loss to compensate for imbalance. Early stopping with a patience of 20 epochs is applied, using validation balanced accuracy as the stopping criterion. Hyperparameters were selected based on preliminary experiments, balancing performance and stability across both benchmark and biological datasets.

