# OpenReview forum: "Mitigating Class Imbalance in Graph-Structured Data via Hierarchical Learning: Insights from Protein Binding Site Prediction"
_ICLR.cc/2026/Conference — Submitted to ICLR 2026_

### Official Review · Reviewer_yENd · 2025-10-30

**Soundness:** 2
**Presentation:** 1
**Contribution:** 2
**Rating:** 2
**Confidence:** 3

**Summary:**

This paper studies the imbalanced node classification problem. To be specific, its workflow is
1. dividing a given into subgraphs
2. predicting whether a subgraph includes at least one minority-class node
3. predicting node labels on subgraphs which are likely to include at least one minority-class node.

**Strengths:**

S1. This paper includes experiments on standard graph benchmarks, Cora, Pubmed, and Citeseer.

S2. This paper includes studies on the biological settings, COACH100 dataset.

**Weaknesses:**

W1. The writing of this paper can be improved. The whole methodology section is based on plain text, which is vague.

W2. The idea of "predicting whether a subgraph includes at least one minority-class node" does not makes sense. The paper mentioned that "Subgraph-Level Classifier" is "trained independently with ground-truth labels at the subgraph level". If I understand correctly, the subgraphs which does not include minority nodes, will be marked 0 in the training data, and will always predict 0 in the inference time; however, it does not make sense as those subgraphs with no labelled minority nodes might also have minority nodes.

**Questions:**

Please check the weaknesses I mentioned.

---

### Official Review · Reviewer_mmx4 · 2025-10-30

**Soundness:** 2
**Presentation:** 2
**Contribution:** 2
**Rating:** 2
**Confidence:** 3

**Summary:**

This paper introduces CLARA, a hierarchical framework designed to address class imbalance in graph-structured data, particularly when minority classes are structurally clustered. The method decomposes node classification into two stages: first, a coarse, subgraph-level classifier identifies and filters graph regions likely to contain minority nodes. Second, a fine-grained node-level classifier operates only within these selected subgraphs to produce the final predictions.

**Strengths:**

S1 (motivation): The paper addresses the problem of class imbalance where minority nodes are structurally clustered.

S2 (intuition): The coarse-to-fine hierarchical approach is intuitive.

S3 (flexibility): The framework flexibility allows the integration of domain-specific heuristics, such as solvent-accessible surface area (SASA) for subgraph generation.

**Weaknesses:**

W1 (limited method details): The proposed method (Section 3) is described only at a very high level, almost in pure text. Crucial technical details such as the GNN architecture, the pooling function ($\rho$) 13, and the loss functions are not introduced, making the core contribution difficult to evaluate or reproduce from the main paper.

W2 (limited evaluation): The evaluation was rather limited. Regarding general node classification, the method was evaluated only on cora, citeseer, and pubmed with synthetic imbalance, but the paper does not use common class-imbalance datasets. Regarding binding site prediction, the method was evaluated only on COACH100, which is small and has limited scope. It is unclear whether the prposed method generalizes to other scenarios.

W3 (unsupported claims): The paper repeatedly claims improved scalability but provides no theoretical or empirical analysis to support this.

W4 (over-reliance on heuristics): The paper's best results are achieved by CLARA-SASA, which relies on a strong, domain-specific heuristic (SASA). The general-purpose version (CLARA with random coloring) performs significantly worse.

**Questions:**

See weaknesses.

---

### Official Review · Reviewer_Bq3G · 2025-10-31

**Soundness:** 2
**Presentation:** 3
**Contribution:** 2
**Rating:** 2
**Confidence:** 3

**Summary:**

This paper proposes CLARA (Coarse-to-fine Localized Adaptive Region Attention), a two-stage method for mitigating class imbalance in node classification on graphs. Stage 1 retrieves subgraphs that are likely to contain minority instances; Stage 2 then identifies minority instances within those subgraphs. The approach is evaluated on three graph benchmarks and a protein–ligand binding dataset.

**Strengths:**

1. The paper evaluates the method on both general graph benchmarks and a specialized biological application (protein-ligand binding), demonstrating broader applicability.

2. The paper is well-written and easy to follow. The hierarchical framework is intuitive and well-illustrated.

**Weaknesses:**

1. In the introduction, the authors claim that "minority instances cluster in compact, topologically meaningful subregions." However, no empirical analysis is provided to validate this assumption on the datasets used. Including a case study demonstrating that minority nodes indeed exhibit spatial clustering in the datasets used would make this claim more convincing. Also, how does the method perform when minority nodes are uniformly distributed rather than clustered?

2. Subgraph Construction:
- The k for k-hop neighborhoods appears to be a critical hyperparameter, but guidance is missing. The choice of k-hop neighborhoods (k=3) appears arbitrary. A sensitivity analysis on k is necessary to understand the method's robustness to this critical hyperparameter.

- What is the class distribution within the extracted subgraphs? If subgraph labels remain highly imbalanced, how does Stage 2 benefit from decomposition? Please report imbalance ratios at the subgraph level. Additionally, what is the classification performance of Stage 1? How many subgraphs are filtered out, and how many subgraphs containing minority nodes are incorrectly discarded?

3. The paper states that CLARA-S uses "subgraph decomposition but omits hierarchical filtering." However, I may be misunderstanding the setup. Could you please clarify how CLARA-S is different from a standard GNN applied to each subgraph separately instead of the full graph? If CLARA-S is indeed a vanilla GNN trained on subgraphs without any special imbalance-handling techniques, it is a bit surprising that it consistently outperforms all baseline methods by a margin in Figure 2, as these baselines are specifically designed to address class imbalance. Could you help explain where the improvement comes from?

4. Experiment 1 (Section 4.1) uses an imbalance ratio of 10, which is relatively moderate. Please evaluate performance at higher imbalance ratios (20, 50, 100) to better understand scalability to severe imbalance scenarios common in real-world applications. Also, how are minority and majority classes defined under the multi-class classification setting? Additionally, Table 1 lacks error bars or confidence intervals like Figure 2, which makes it difficult to assess statistical significance of the results.

5. Source code is not provided for reproducibility.

**Questions:**

See Weakness

---

### Official Review · Reviewer_feaF · 2025-11-02

**Soundness:** 2
**Presentation:** 2
**Contribution:** 1
**Rating:** 2
**Confidence:** 4

**Summary:**

This paper proposes CLARA, a hierarchical coarse-to-fine framework for handling class imbalance in GNNs. The key idea is to decompose node classification into two stages. The authors motivate their work by arguing that existing methods  fail to address scenarios where minority nodes form topologically compact clusters, which is common in domains like bioinformatics.

**Strengths:**

1. The idea of using a coarse subgraph-level filter to focus computational resources and model capacity on promising regions is intuitive and novel for the graph imbalance problem.

2. This paper is easy to understand.

**Weaknesses:**

1. As noted, the experimental section is critically underdeveloped. The claims of "broad and consistent effectiveness" and "state-of-the-art results" are not sufficiently supported.

2. It is unclear how much each component (the coarse classifier, the fine-grained predictor, the heuristic guidance) contributes to the overall performance. Without ablation, the necessity of the full framework is not proven.

3. Lack of theoretical or in-depth analysis.

**Questions:**

1. The method seems heavily inspired by and tailored for biological networks where minority classes form compact clusters. How well does it perform on general graph datasets where minority nodes might be more diffusely distributed? The coarse filter might struggle or provide little benefit in such scenarios.

2. While the framework is touted as scalable, what is the end-to-end computational cost? The process of generating candidate subgraphs and then running a two-stage model could be more expensive than a single pass over the graph, especially if the coarse classifier has a high false positive rate, leading to many subgraphs being processed unnecessarily.

---

### Meta-Review · Area_Chair_VxtF · 2026-01-06

**Summary:**

The reviewers unanimously identified major shortcomings in the paper. Key concerns include critically underdeveloped experiments, a vague methodological description lacking essential details, and unvalidated core assumptions. The novelty and effectiveness of the proposed framework are not sufficiently demonstrated, and major claims such as scalability are unsupported. Consequently, all reviewers recommend rejection in the current form.

**Reviewer Concerns:**

no rebuttal

**Reviewer Scores:**

no rebuttal

---

### Decision · Program_Chairs · 2026-01-26

Reject